# Intercomparison Experiment of Water-Insoluble Carbonaceous Particles in Snow in a High-Mountain Environment (1598 m a.s.l.)

Outi Meinander [1,*], Anne Kasper-Giebl [2], Silvia Becagli [3], Minna Aurela [1], Daniela Kau [2], Giulia Calzolai [4] and Wolfgang Schöner [5]

1. Finnish Meteorological Institute, Aerosols and Climate, 00560 Helsinki, Finland; minna.aurela@fmi.fi
2. Institute of Chemical Technologies and Analytics, TU Wien, 1060 Wien, Austria; anneliese.kasper-giebl@tuwien.ac.at (A.K.-G.); daniela.kau@tuwien.ac.at (D.K.)
3. Department of Chemistry "Ugo Schiff", University of Florence, 50019 Florence, Italy; silvia.becagli@unifi.it
4. National Institute for Nuclear Physics, Florence Division, Sesto Fiorentino, 50019 Florence, Italy; calzolai@fi.infn.it
5. Department of Geography and Regional Sciences, Karl-Franzens-Universität Graz, 8010 Graz, Austria; wolfgang.schoener@uni-graz.at
* Correspondence: outi.meinander@fmi.fi

**Abstract:** The harmonization of sampling, sample preparation and laboratory analysis methods to detect carbon compounds in snow requires detailed documentation of those methods and their uncertainties. Moreover, intercomparison experiments are needed to reveal differences and quantify the uncertainties further. Here, we document our sampling, filtering, and analysis protocols used in the intercomparison experiment from three laboratories to detect water-insoluble carbon in seasonal surface snow in the high-mountain environment at Kolm Saigurn (47.067842° N, 12.98394° E, alt 1598 m a.s.l.), Austria. The participating laboratories were TU Wien (Austria), the University of Florence (Italy), and the Finnish Meteorological Institute (Finland). For the carbon analysis, the NIOSH5040 and EUSAAR2 protocols of the OCEC thermal-optical method were used. The median of the measured concentrations of total carbon (TC) was 323 ppb, organic carbon (OC) 308 ppb, and elemental carbon (EC) 16 ppb. The methods and protocols used in this experiment did not reveal large differences between the laboratories, and the TC, OC, and EC values of four inter-comparison locations, five meters apart, did not show meter-scale horizontal variability in surface snow. The results suggest that the presented methods are applicable for future research and monitoring of carbonaceous particles in snow. Moreover, a recommendation on the key parameters that an intercomparison experiment participant should be asked for is presented to help future investigations on carbonaceous particles in snow. The work contributes to the harmonization of the methods for measuring the snow chemistry of seasonal snow deposited on the ground.

**Keywords:** snow; carbon; black carbon; elemental carbon; organic carbon; brown carbon; total carbon; intercomparison; thermal-optical method; protocol

## 1. Introduction

Wet and dry deposition of light-absorbing atmospheric aerosols, such as black carbon (BC) or brown carbon (BrC, a component of organic carbon, OC), darken bright surfaces, and can enhance or initiate snowmelt. Black carbon is a short-lived climate forcer (SLCF) that undergoes regional and intercontinental transport from source regions during its short atmospheric lifetime [1,2]. BC (or thermally determined elemental carbon (EC), a close surrogate of BC) originates mainly from incomplete combustion of carbonaceous materials such as biomass or fossil fuels [3]. For simplicity, the term BC also refers to EC in this paper. Organic carbon is co-emitted with BC or may be released from soils or be formed from

gaseous precursors, i.e., volatile organic carbon compounds (VOC). Carbon compounds in snow are also part of the global carbon cycle.

There are no standardized methods or protocols for sampling, pre-treatment and analysis of carbonaceous particulate matter in snow. Earlier, protocols for snow sampling and for snow chemistry, e.g., to detect BC content in snow, were presented in [4–6]. The protocols and recommendations for sampling snow for BC in Svalbard seasonal snowpack on glaciers were included in [4], where the term protocol referred to sampling only. The snowpack chemistry monitoring protocol for the Rocky Mountain Network, USA, was presented in [5], where the term protocol included snowpack sampling procedures, discussion about optimal sample spacing, and a detailed set of standard operating procedures for annual snowpack sampling. Hence, the protocol in [5] referred to sampling for snow chemistry, including dissolved organic carbon but did not mention water-insoluble carbon. The sampling protocol was claimed to be adaptable for the collection of samples for selected organic compounds. Sampling, filtering, and analysis protocols to detect BC, OC, and TC (total carbon, defined as the sum of BC and OC) in seasonal surface snow in an urban background and Arctic Finland (>60° N) were given in [6], where the term protocol referred to the analysis protocols of the OCEC thermal-optical method. The OCEC method is a European standard for atmospheric EC, used and investigated in [6] for snow samples. In [5], the terms method and procedure were used for sampling and filtering. In addition to [4–6], methods used for measuring carbon in snow, in various environments, were presented as part of research papers not dedicated to protocols, e.g., in [7–11]. The protocols of [4–6] follow in more detail.

The protocols for snow sampling in Svalbard of seasonal snow on a glacier [4] were designed to focus on snow–atmosphere interactions and to exclude the effects of soil–snow interactions. In addition to sampling, the technical report [4] discusses the best snow sampling strategy and contamination risks during sampling. No literature references, instructions or recommendations for filtering or chemical analysis of BC, i.e., actually referring to EC regarding the analytical method, are given there. For sampling, [4] presents sampling equipment including, e.g., sterile Whirl-Pak bags and sterilized Teflon or stainless-steel sampling tools. They recommend approximately 10 L of fresh and light snow per sample and 5 L of denser or old snow (melt refreeze crust). This is based on their experience, which shows that ~2–3 L of melt water volume is needed to measure BC in Svalbard snow using the thermal-optical OCEC method. The sampling options in [4] include sampling by fixed-depth increment and sampling per discrete snow layers. The latter one has the potential benefit of linking a snow layer and its properties to a specific climate event, such as a period of precipitation or surface melt or tracking the origin of air associated with specific snow accumulation periods. It also enables the correlation of intervals of snow accumulation between separate snow pits at different altitudes, such as that on a glacier between its ablation and accumulation area. The reasoning behind this is that the snowpack stratigraphy may vary over an altitude range with the precipitation amount and phase (rain vs. snow) so that the same precipitation event might be represented by an ice layer in the lower reaches of a glacier and by a snow layer in its upper part [4]. Hence, sampling per discrete layer allows connecting and comparing snowpack properties in different parts of the same glacier. If repeated sampling is planned at a given site, they say that sampling by discrete layer allows the tracking of how surface melting modifies the physical properties of the snow. Their first sample is taken from the top 5 cm of the snowpack as BC in the uppermost layers has the largest impact on snow albedo. The sampling is then continued at approximately 50 cm depth increments down to the bottom of the snow pit, and if the lowermost sample is less than 10 cm thick, it is combined with the one above it. The authors state that sampling by fixed depth increments is the simplest and easiest protocol to use by multiple teams in the field, since a priori description of the snowpack stratigraphy is not needed. If it is of interest to know exactly how snowpack properties vary as a function of depth, a fixed-depth interval strategy is the best for comparing results from multiple snow pits, and temporal variations can be compared without accounting for precipitation events.

They also say that sampling by fixed-depth increment is particularly useful when studying the radiative properties of snow and photochemical processes in the snowpack and also for quantifying the deposition of atmospheric species on the snow surface with the focus on snow–atmosphere interactions. Hence, they conclude that both strategies have advantages and inconveniences, and the choice must be guided by the research objectives and needs.

The snowpack chemistry monitoring protocol for the Rocky Mountain Network (ROMN), USA, was presented in [5], which included snowpack sampling procedures, a discussion about optimal sample spacing based upon snowpack samples, and detailed standard operating procedures (SOPs) for annual snowpack sampling and pre- and post-season preparations. The SOPs describe advance planning, collection-permit compliance, equipment preparation, personnel training, safety considerations, sampling-site operations, quality assurance, sample handling and analyses, and data management and reporting. Their analytical methods include pre-processing snowmelt samples and delivering samples to the selected laboratories for analyses, laboratory methods and quality assurance, and acceptable detection limits. Their monitoring protocol includes dissolved organic carbon and selected organic compounds, but it does not include water insoluble carbonaceous particles or BC. For [5], the chemical analyses of snowpack samples are indicators of recent air quality conditions, and the key objective was to optimize the regional representation of atmospheric deposition. Their protocol has also been applied in other mid-latitude and high-latitude mountain ranges in the Western United States and Alaska. They state that although seasonal snowpacks do not represent several months of summertime precipitation, the snowpack sampling methodology enables efficient collection of a substantial fraction of annual precipitation in a single sample, and as annual snowpacks melt, atmospheric inputs of these compounds to the watersheds may affect aquatic and terrestrial ecosystems. According to [5], the coverage for high-elevation areas, i.e., greater than 2400 m, in the Rocky Mountains is limited, although high-elevation snowpacks are important because they accumulate two to three times the annual precipitation measured at lower elevations, where regular monitoring is more easily accomplished. They say that the importance of investigating the snowpack chemistry in the Rocky Mountains comes from the snowfall, which accumulates from October until March, April, or May, and provides about 50–70% of the annual precipitation in the headwater basins of the Rocky Mountains. These annual snowpacks collect both wet and dry deposition and provide a record of the deposition of airborne contaminants until snowmelt begins each spring. As snowmelt supplies most of the freshwater in mountain lakes, streams, and wetlands, they monitor the water quality of snow to understand the effects of atmospheric deposition on these systems. They conclude that revising their protocol may be necessary as circumstances change through time or as errors are discovered.

The methods for sampling and filtering, as well as analysis protocols to detect BC, OC, and TC in seasonal surface snow in an urban background environment and in the European Arctic north of 60° N in Finland, were presented in detail in [6]. There, surface snow, defined as the top 2–3 cm, was collected. The supplementary material of [6] shows the details of the method using multiple photographs. Some of the error sources are also studied and quantified in [6], including, e.g., if carbon could remain attached to the sampling bag, the influence of non-homogenous snow samples on the detected concentrations, and the leaking of filters as a result of the large amounts of water filtered through. In addition, benefits and drawbacks of various methods for BC and OC in snow are listed, based on their experience of using OCEC, SP2, optical methods, and gravimetry [6]. They say that to reveal the origin of the impurities in snow using long-range transport modeling, the sample collection area, the exact time of the sample collection, and whether the sampling was within 24 h of the latest snowfall need to be known. For studies on melting snow, they state that it is necessary to know whether impurities disturb the water-holding capacity of melting snow and to have information on surface density according to the sampled snow layer. They continue that for radiative transfer (RT) modeling, the values for snow grain size, snowpack thickness/depth, and snowpack density (snowball test) are

most often needed, and if RT modeling is used to verify albedo measurement, knowledge of cloud conditions is essential, too, and clear or fully cloudy are the best cloud conditions for this purpose. They conclude that the aim of sampling in detecting carbon in snow is to measure atmospheric carbon deposition in time or space (or both) and to understand post-depositional cryospheric processes. The time can be from seconds to seasons or long-term monitoring over years or decades. Space includes spatial variability, for example, in meter scale, as well as horizontal sampling (e.g., sampling every 10 m in rows, grids, or randomly) and vertical sampling (e.g., surface, layers, bulk, or fixed centimeters). According to [6], the work benefits from including measurements on ancillary information, and the environmental parameters relevant for carbon in snow studies depend mostly on the aim of the study. For example, ancillary measurements can include snow depth and melt periods (to detect the impurity accumulation and surface enrichment), snow albedo (to detect radiative effects), and precipitation (to detect the origin of the impurities). They state that quality assurance (QA) and quality control (QC) as well as the identification and quantification of the uncertainties (where repetition of the measurement produce a result within an interval around the measured value) and of the possible error sources (referring to disagreement between a measurement and the true/accepted value) of the measured carbon data should be included in the carbon detection results. Finally, the data should be reported correctly. In [6], the distribution of BC in snow values was found to be positively skewed (number of samples $n = 107$, skewness $\gamma_1 = 0.12$), and carbon results were thus reported as median values instead of as average and standard deviation, which are valid for normally distributed data.

Harmonization of sampling, sample preparation, and analysis methods to detect water insoluble carbonaceous particles in snow requires detailed documentation of the methods and their uncertainties. Moreover, intercomparison experiments are needed to reveal potential differences and error sources. They can also help to identify and quantify uncertainties further. Currently, lack of suitable certified reference material to analyze OC, BC, and TC in snow makes such a laboratory intercomparison impossible. Furthermore, as [4–6] demonstrate, snow sampling is an essential part for snow chemistry intercomparisons. Snow cover can be sampled using bulk samples, single layer samples, or surface snow samples. This intercomparison study focuses on surface snow, determined as the first 1–2 cm of snow surface, and us a method where snow samples are melted and filtered prior to the chemical analysis. Similar to [6], the term protocol refers here to the OCEC analysis protocols, while for sampling and filtering the terms method and procedure are used. Procedure can be understood as a way of doing something in a correct way, while method would refer simply to how the work was conducted. In [4,5], the term procedure was used for snow sampling and can be understood to refer to a plan for performing a scientific experiment, while in [6] it was used to refer to the chemical analysis protocols. The thermal-optical OCEC method, which is used here for snow samples, is the current European standard for determining atmospheric EC [12,13], and in the atmospheric OCEC intercomparisons, it has been found that results are worse with smaller TC contents [14]. The focus of this investigation is on carbon compounds in snow which originate from wet or dry deposited atmospheric aerosols, i.e., snow–atmosphere interactions. The main aim of this work was to document the methods and investigate the differences of three different laboratories to detect water-insoluble OC, BC, and TC concentrations from snow samples in a remote high-mountain environment. The authors are not aware of any other previous field intercomparison experiments from multiple laboratories on the topic of carbon in snow. The experiment also revealed the concentrations of the various carbon compound concentrations in surface snow and their meter-scale variability at the time of the experiment. As an outcome of the experiment, the key parameters recommended to be asked from an intercomparison participant prior the experiment were listed for future inter-comparisons. The work contributed to the efforts on harmonization of the methods in snow chemistry.

## 2. Materials and Methods

### 2.1. Site Description

Snow samples were collected and analyzed from the high-mountain environment at Kolm Saigurn (47.067842° N, 12.98394° E, alt 1598 m), Austria (Figure 1). The sampling site at Kolm Saigurn is located below the treeline in an Alpine pasture landscape, in the foot of the Mountain Sonnblick with Sonnblick Observatory at the summit. Sonnblick Observatory is part of the Global Atmosphere Watch (GAW), Baseline Surface Radiation Network (BSRN), and Global Cryospheric Watch (GCW) networks of the World Meteorological Organization (WMO), and monitors data with the focus on cryosphere, permafrost, biology, meteorology, air chemistry, aerosols, snow chemistry, and deposition. The inter-comparison experiment on carbon in snow was part of the Workshop on "Integrated long-term snow chemistry monitoring", on 26 February–2 March 2018, of the European Union EU ESSEM COST Action ES1404 Harmosnow-project of "A European network for a harmonized monitoring of snow for the benefit of climate change scenarios, hydrology, and numerical weather prediction". In total, 20 researchers from eight European countries participated at the workshop to exchange their experiences and results on measuring chemical properties of the snow cover. Three researchers from three laboratories (TU Wien, Austria, the University of Florence, Italy, and the Finnish Meteorological Institute (FMI), Finland) participated in the carbon compound intercomparison.

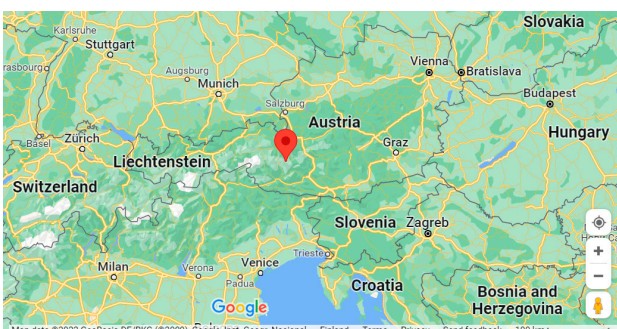 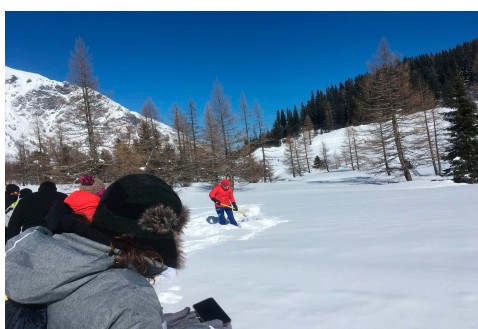

**Figure 1.** The sampling site at Kolm Saigurn, Austria. (**Left**) Map of the study location (unit distance of 100 km; Google Maps). (**Right**) Photo of the sampling area.

### 2.2. Snow Sampling

The snow samples were collected in a location with open flat terrain at the end of the Rauriser Valley (Hüttwinkl Valley) at the foot of the Rauriser Sonnblick (3106 m). The time of sampling on 28 February 2018 was characterized by an approximately 170 cm winter snow cover with a dry and cold (between −15 °C and −5 °C) snow layer on top that was only weakly metamorphosed. The weather during sampling was cold (below 0 °C), without wind (calm), and with bright sunshine. The samples for carbon compound analysis were collected from undisturbed surface snow by collecting the top two centimeters of snow. Stainless steel sampling spoons and sterile sampling bags were used. The snow sampling was several meters away from an area with four snow pits, which were used for other chemistry intercomparisons (including major ion concentrations and stable isotopes) and to measure physical properties of the snow cover, e.g., snow temperature and density profiles.

First, the sampling for carbon detection for the intercomparison experiment of three laboratories was designed. The sampling procedure was as follows. Four sampling locations were used. The sampled area in sampling spot number one, Spot 1 (S1), was approximately one square meter, measured with the ruler on the snow surface (Figure 2). The sampling areas of Spot 2 (S2), Spot 3 (S3), and Spot 4 (S4) were approximately two square meters each. In each location, the first spoon of snow was placed in "bag1" of laboratory number 1, then the second spoon to "bag2" of laboratory number 2, and the third spoon to "bag3" of laboratory number 3. Then, the second-round sampling of new untouched surface snow in the same sampling location continued next to the already sampled surface snow and

resulted in one spoon again in bag1, then one in bag2, and one in bag3, etc., until the sampling area of each spot was covered. As a result, four independent and as-homogenous-as-possible surface-snow samples for each laboratory were gained, and any potential small-scale (less than one meter scale) differences in carbon compound concentrations in surface snow were aimed to be eliminated. In addition, extra bags from the sampling spots S1–S4 were collected for TU Wien as follows: one bag from S2, one bag from S3 and three bags from S4. This resulted in a total of nine snow sample bags for TU Wien, 4 sample bags for the University of Florence and four sample bags for Finnish Meteorological Institute. As a result, the total number of snow samples of the experiment was 17. Two locations (S1 and S2) were on the West side of the snow pits (used for other inter-comparisons) and in approximately 3–5 m distance from each other, while two other locations were on the East side of the four snow pits (S3 and S4), approximately 3–5 m distance from each other (Figure 3). The longest distance from one location to another (from S2 to S4) was approximately 30–50 m. With the help of sampling sites at both ends of the sampling line, it was possible to detect both meter- and tens-of-meters-scale horizontal differences in surface carbon.

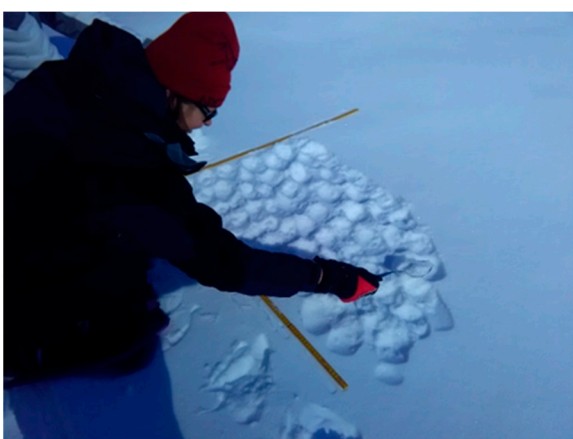

**Figure 2.** Snow sampling for the laboratory intercomparison at Kolm Saigurn, Austria.

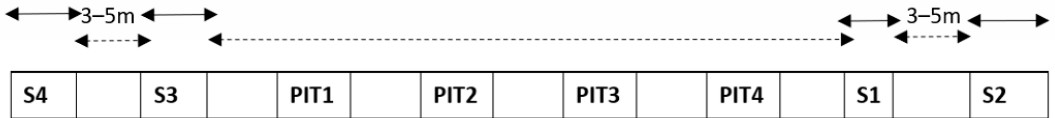

**Figure 3.** The sampling locations of the carbon in snow experiment (Spots S1–S4), and the locations of the four snow pits (Pits 1–4) used for other snow chemistry studies in Kolm Saigurn, Austria. The width of the sampling spot number 1 (S1) was one meter. The width of the sampling spots 2–4 (S2–S4) was two meters. The space between the sampling spots was approximately 3–5 m. The distance from S2 to S4 was approximately 30–50 m.

### 2.3. Sample Storage, Melting and Filtering

The snow samples for University of Florence and Finnish Meteorological Institute were melted and filtered on the day of sampling, while the samples for TU Wien were stored frozen until used in the laboratory. Hence, the samples for Italy and Finland were transported as filters and TU Wien as frozen snow.

#### 2.3.1. TU Wien

Quartz fiber filters (PALLFLEX® Tissuquartz™, Pall Laboratory, New York, NY, USA) with a diameter of 27 mm were baked before the filtration at 600 °C for 24 h and cooled in a desiccator above distilled water for 24 h. All glassware was rinsed with Milli-Q water before usage. In the laboratory, the snow samples were melted in a glass beaker using a microwave (approximately 3 times 1 min at 600 W). Samples were weighed before and

after melting in the microwave and the difference was below 0.1%, so practically no loss of water occurred during melting. Filtration onto the quartz fiber filters was performed by applying a slight vacuum (low pressure of 0.3 to 0.6 bar). The beaker was rinsed with Milli-Q water to ensure quantitative transfer of the sample. The filters, with a circular loaded area of 16 mm diameter, were dried above silica gel in a desiccator at room temperature overnight. To allow analysis with different protocols and to load the OCEC instrument properly, a punch with a diameter of 15 mm was made and cut into two halves subsequent to drying. Both halves were weighed to correct for irregular partitioning. As the punch of 15 mm diameter was smaller than the loaded area, a correction factor was applied to account for the outermost part of the loaded filter. This factor was determined by analyzing the remains of the loaded area, where some enrichment of particles was visible. As analysis of the filters was performed on the day after filtration, and no storage at lower temperature prior to analysis was necessary.

### 2.3.2. University of Florence

The filtration system in Italy was Sartorius™ Polycarbonate Filtration Holders (commercialized by fisher scientific, https://www.fishersci.co.uk/shop/products/sartorius-sartorius-polycarbonate-filtration-holders-2/p-4231980 (accessed on 20 March 2022). CHM Chem Lab Group (Barcelona, Spain) pre-fired quartz filters were used to collect insoluble TC particles in the melted snow. Each 47 mm diameter quartz filter was punched in a 25 mm diameter dish (Figure 4), where the melted snow (typically 300 mL) was passed through. The remaining part was used for blank measurements. The filtering equipment was not washed during the filtering process.

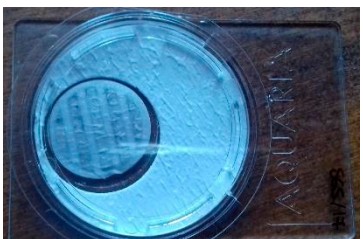

**Figure 4.** Example of filter sampled at University of Florence for insoluble EC/OC/TC measurements in melted snow.

### 2.3.3. Finnish Meteorological Institute

The quartz fiber filters (model T293, Munktell Filter AB, Falun, Sweden) with a diameter of 47 mm had been pre-burned at FMI in Finland for 4 h at 800 °C before the intercomparison experiment. Snow samples were melted on the day of sampling and filtered using a hand pump system attached to the filtering system to create a vacuum during filtering. The filtering equipment was washed during the filtering process utilizing the filtration water. The filter area where the melted snow was passed through was 13.2 cm$^2$, i.e., the whole filter area except the very edges that were under the glass filtering cup surrounding the filter.

### *2.4. OCEC Analysis*

In the participating laboratories, the filter samples were analyzed using a thermal-optical carbon analyzer (OCEC Aerosol Analyzer, Sunset Laboratory, Tigard, OR, USA, model 5 L). The thermal-optical OCEC method was created by [15], where a more detailed description of OCEC is given. The samples were analyzed using the protocols of EUSAAR2 (Austria, Finland) and NIOSH5040 (Austria, Italy). In TU Wien laboratory analysis, one half of each filter was analyzed using the EUSAAR2 protocol. The same protocol was used by FMI in this intercomparison experiment. The other half was analyzed with the NIOSH5040 protocol. That protocol was the same as used by University of Florence for the intercomparison. In University of Florence analysis, as the deposited particles were not homogeneously

distributed over the filter, the 25 mm dish was divided in four parts and completely analyzed. OC and EC concentration are obtained by summing the concentrations in each portion of the filter. In Finnish Meteorological Institute the filters were analyzed using the EUSAAR2 protocol and three replicates were analyzed for each sample.

The basic principle of the OCEC analysis is as follows. A filter piece or an aliquot of a filter is put inside the sample oven and is then heated in two phases so that the carbon on the filter is oxidized to carbon dioxide ($CO_2$). First, a helium atmosphere is used to oxidize OC into $CO_2$. Secondly, an oxygen–helium mixture is used to vaporize and oxidize the EC into $CO_2$. Carbon dioxide is reduced to methane, which is measured using a flame ionization detector (FID). The split point, which separates OC and EC, is determined using laser transmittance. This is needed since during the first phase, some of the OC is pyrolized, and it is released during the second phase. During pyrolysis, the laser transmittance signal decreases. At the split point, the laser transmittance signal returns to its original value.

The automatic split point was used for EUSAAR2, as the setting of the initial laser transmission did not show any special features. The slope of the laser transmission around the split point was very high and even minimal changes would lead to a discrepancy with the initial laser transmission signal. The automatic split point was set in the peak of the EC3 temperature step for all samples. For the measurements with the NIOSH 5040 temperature protocol, the automatic split point was used as well. Again, the change of the laser transmission around the split point showed a steep slope. The automatic split point was set in the EC2 or the EC3 temperature step.

A blank and a sucrose standard was run every day before analysis of the samples. Sucrose measurements were performed as a procedure for using the instrument. These values can be used to estimate the standard deviation of the analysis. At FMI and University of Florence, the volume of meltwater was used for concentration conversions into ng/L, which equals parts per billion by mass (ppb), which is equal to µg/kg, determined as µg-C/L-$H_2O$. At TU Wien, the sample weight was used for concentration conversions determined as µg-C/kg. As the density of water for the temperatures typical for indoor laboratories is close to 1 kg/L (e.g., 0.99802 at 21 °C), this is assumed to have no influence on the results. The uncertainty of the OCEC method is 5% relative error for higher loaded samples, according to Sunset Laboratory, Inc.

The total number of filter sample analyses performed was 34 sample filter analyses for OC, EC, and TC, and additional sucrose and blank filter analyses. TU Wien had 18 sample filter analyses, University of Florence had 4 sample filter analyses (accompanied with 4 blank filters analyzed separately for each sample filter), and Finnish Meteorological Institute had 12 filter sample analyses (using 3 replicates for each sample filter). This resulted in a total of 102 carbon-in-snow sample analysis results: 34 for OC, 34 for EC, and 34 for TC, which were all used to investigate the differences between the three laboratories and the amounts and variability in carbon in snow in meter- and ten-meter-scales.

*2.5. Summary of the Differences in Methods of the Intercomparison*

For all the intercomparison samples, the sampling was made using the same sampling method, meaning that there were three sets of samples which could be assumed homogeneous. Differences between the laboratories occurred in storage, melting, and filtering. For filtering, each laboratory used filters of different sizes. Analysis method was the same OCEC thermal-optical method, where sucrose tests were performed to reveal potential differences between the instruments. The EUSAAR2 protocol was used in Finland, the NIOSH5040 protocol in Italy, and both protocols in Austria.

**3. Results and Discussion**

Here we present the field observation results of the snow properties at the time of sampling, then laboratory analysis results of sucrose and blank filter measurements, and the results of OC, EC, and TC concentrations in the filter samples from the three participating

laboratories. This is followed by a discussion on the discrepancies found between the laboratories and in the concentrations detected in snow.

### 3.1. Snow Properties

The total snow depth at the time of the intercomparison experiment was 170 cm, with an upper dry-snow layer of approximately 90 cm thickness. The lower layer had a clear structure of various metamorphism processes including thin ice layers near to the ground. The snow surface temperature was <0 °C.

### 3.2. Sucrose Measurements

At TU Wien, the concentration of the sucrose solution used was 5 gC/l. The results of a 10 μL sample measured at the start of each measurement day was 49.5; 51.1, and 50.6 μgC/cm$^2$, respectively. Considering the 5% uncertainty of the analyzer, these results are not distinguishable from the expected 50 μgC/cm$^2$.

At the University of Florence, a complete calibration curve was carried out periodically (about once a month), measuring standard solutions of sucrose at four different concentrations: 0.5, 1, 2, and 4 μg/μL; briefly, 10 μL of each solution was dripped, using an Eppendorf pipette, on a blank filter and then analyzed as the unknown samples. The calibration performed before and after the measurements of snow samples differed by 3.3% and 12% at concentrations of 40 μg/cm$^2$ and 10 μg/cm$^2$, respectively.

At the Finnish Meteorological Institute, the concentration of the sucrose solution used was 5 gC/l. Sucrose was measured at the beginning of each OCEC analysis day. For the intercomparison samples, the prior sucrose analysis value was 27.94 ± 1.5 μgC for the Sunset sucrose standard of 27.28 μgC. Overall, the uncertainty of the OCEC of the Finnish Meteorological Institute was estimated to be ±0.2 μgC (+5% relative error for higher loaded samples) [6].

### 3.3. Blank Filters

At TU Wien, the blank value of filters (diameter 27 mm), which underwent the process of filtration and cutting, was 1.6 μgC/cm$^2$. There was no EC fraction, and therefore the blank value was subtracted only from TC or OC values, respectively.

At the University of Florence, the filter OC, EC, and TC blanks were measured on the unsampled part of each 47 mm filter (Figure 4), and therefore each sampled filter (diameter 25 mm) had its own blank. This procedure takes into account the large variability observed in these quartz filters (Caiazzo et al., 2021). Blank values for each analyzed filter are presented in Section 3.4 (as part of the University of Florence OCEC results in Table 1) and these values were subtracted from the respective OC, EC, and TC measurements on the sampled portion. Expressing these values as a carbon load per cm$^2$, they range from 1.6 μgC/cm$^2$ to 2.5 μgC/cm$^2$, which corresponds quite nicely to the lab blank given for TU Wien. The slightly higher numbers can be expected as filtration of the University of Florence samples was performed at the site.

**Table 1.** Results of the University of Florence snow sample analysis.

| Sample ID | Volume of Melted Snow mL | Mass of EC in the Filter μg/Filter (±unc.) | Mass of OC in the Filter μg/Filter (±unc.) | Mass of EC in the Filter Blank μg/Filter (±unc.) | Mass of OC in the Filter Blank μg/Filter (±unc.) | EC per Volume of Snow Ppb (±unc.) | OC per Volume of Snow Ppb (±unc.) | TC per Volume of Snow Ppb (±unc.) |
|---|---|---|---|---|---|---|---|---|
| S1Flo | 300 | 3.84 (±0.80) | 102.3 (±6.2) | 0.00 | 12.4 (±1.2) | 12.8 (±2.7) | 299.9 (±20.5) | 312.8 (±23.1) |
| S2Flo | 300 | 4.95 (±0.86) | 202.2 (±11.6) | 0.23 (±0.57) | 11.2 (±1.2) | 15.7 (±2.9) | 636.8 (±38.8) | 651.1 (±41.5) |
| S3Flo | 300 | 3.48 (±0.77) | 84.5 (±5.2) | 0.00 | 9.6 (±1.1) | 11.6 (±2.6) | 249.5 (±17.2) | 261.1 (±19.8) |
| S4Flo | 330 | 5.76 (±0.89) | 140.6 (±8.3) | 0.00 | 7.6 (±1.0) | 17.4 (±2.7) | 403.1 (±25.0) | 420.6 (±27.7) |

At the Finnish Meteorological Institute, blank filters were not used in this experiment. Instead, three replicates of each sample filter (diameter 47 mm) were analyzed (Section 3.4). Traveling blank filters, which undergo the process of transport and cutting, have elsewhere shown no EC fraction and TC or OC less than 4 µgC/cm$^2$.

### 3.4. OC/EC Results

TU Wien carried out the comparison of the EUSAAR2 and NIOSH5040 analysis protocols. The comparison is based on the entire data set (nine snow samples). All the filters were cut in half to allow independent measurements with the two protocols. The volume of the filtered sample as well as the final results in ppb are shown in Table 2. The additional visualization of the data in Figures 5–7 shows which spots were sampled only once or several times. The uncertainty of the measurements could not be calculated from multiple measurements, since the sample filters were measured only once with either EUSAAR2 or NIOSH5040. Thus, the error bars reflect an uncertainty of 17% for TC and OC and 20% for EC, which is based on an independent set of snow samples, allowing replicate analysis (5 samples). To compare the protocols, the differences between the concentrations determined with EUSAAR2 and NIOSH5040 were calculated for all of the nine samples and related to the respective mean value. Thus, deviations of 5–33% (TC), 5–36% (OC) and 2–45% (EC) were calculated. The respective medians of 12% (TC) and 14% (OC) are slightly below the measurement uncertainty, indicating that no difference between the protocols is evident. In case of EC the deviations between the protocols range higher up and the median (23%) is slightly larger than the measurement uncertainty, but the results are in quite good agreement. However, based on the limited amount of samples, no final answer on the comparability of the protocols can be given. Overall, the NIOSH5040 temperature protocol has a tendency to give lower concentrations than the EUSAAR2.

**Table 2.** Results of the snow samples analyzed at TU Wien using the EUSAAR2 and NIOSH5040 protocols. Uncertainty of 20% added to EC results and 17% uncertainty added to OC and TC results.

| Sample ID | Volume of Melted Snow (mL) | EUSAAR2 | | | NIOSH5040 | | |
|---|---|---|---|---|---|---|---|
| | | EC (ppb) | OC (ppb) | TC (ppb) | EC (ppb) | OC (ppb) | TC (ppb) |
| S1/1 | 289 | 15.7 ± 3.1 | 252 ± 42.9 | 268 ± 45.6 | 15.4 ± 3.0 | 266 ± 45.2 | 281 ± 47.8 |
| S2/1 | 339 | 19.1 ± 3.8 | 237 ± 40.3 | 256 ± 43.6 | 17.6 ± 3.5 | 340 ± 57.9 | 358 ± 60.9 |
| S2/2 | 298 | 14.7 ± 2.9 | 279 ± 47.5 | 294 ± 50.0 | 14.4 ± 2.9 | 306 ± 51.9 | 320 ± 54.4 |
| S3/1 | 269 | 27.8 ± 5.6 | 489 ± 83.1 | 517 ± 87.8 | 21.3 ± 4.3 | 539 ± 91.6 | 552 ± 93.8 |
| S3/2 | 310 | 14.9 ± 3.0 | 449 ± 76.3 | 464 ± 78.8 | 11.2 ± 2.2 | 416 ± 70.7 | 427 ± 72.6 |
| S4/1 | 370 | 30.4 ± 6.1 | 382 ± 64.9 | 412 ± 70.0 | 19.2 ± 3.8 | 327 ± 55.5 | 346 ± 58.8 |
| S4/2 | 211 | 18.4 ± 3.7 | 302 ± 51.4 | 321 ± 54.5 | 13.2 ± 2.6 | 347 ± 59.0 | 360 ± 61.2 |
| S4/3 | 512 | 23.8 ± 4.8 | 341 ± 58.0 | 365 ± 62.1 | 29.9 ± 6.0 | 440 ± 74.8 | 470 ± 79.9 |
| S4/4 | 491 | 28.4 ± 5.7 | 380 ± 64.6 | 409 ± 69.5 | 24.8 ± 5.0 | 323 ± 54.9 | 348 ± 59.1 |

Furthermore, the analysis at TU Wien allows us to evaluate differences between the samples collected within one sampling spot. The average concentrations of the four bags taken in spot 4 were 376 (TC), 351 (OC), and 25.3 (EC) ppb when analyzed with EUSAAR2, and 381 (TC), 359 (OC), and 21.8 (EC) ppb when analyzed with NIOSH5040. These average values are in quite good agreement with the single measurements when the measurement results and uncertainty ranges are considered. Still, some differences remain, which might point to inhomogeneities during sampling. Regarding EUSAAR2, one sample (S4/2: TC, EC) gave concentrations discernible from the average, while two samples showed deviations for NIOSH5040 (S4/3: TC, OC, EC; S4/2: EC). When the duplicate sampling for spot 2 and 3 is considered, deviations are only found for EC. This again points to the fact that

the reproducibility of sampling and analysis of EC is most critical. Overall, the sampling procedure seems to be well suitable to obtaining representative snow samples.

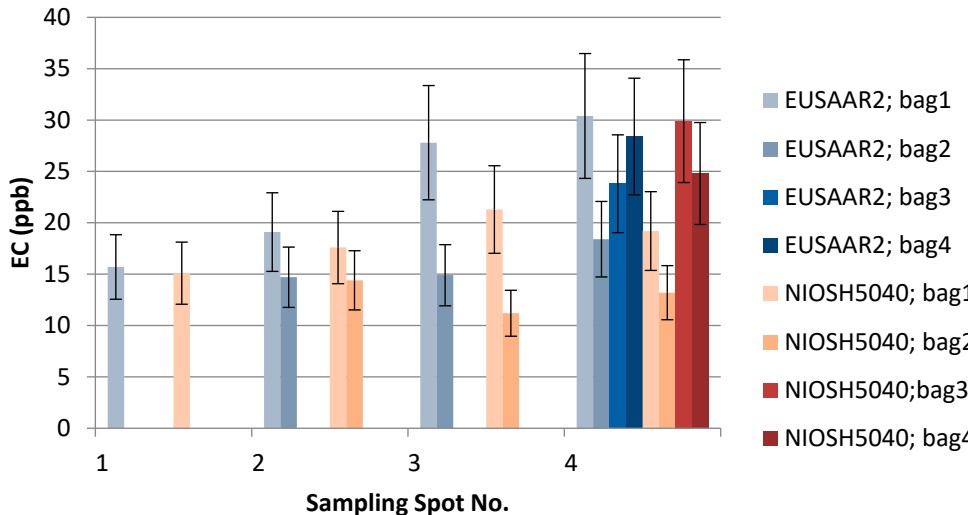

**Figure 5.** The effect of EUSAAR2 and NIOSH5040 analysis protocols on the detected EC.

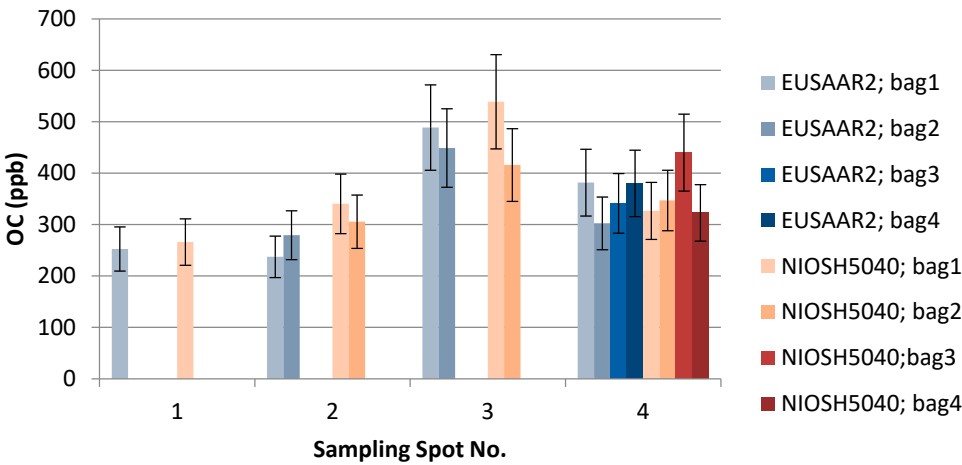

**Figure 6.** The effect of EUSAAR2 and NIOSH5040 analysis protocols on the detected OC.

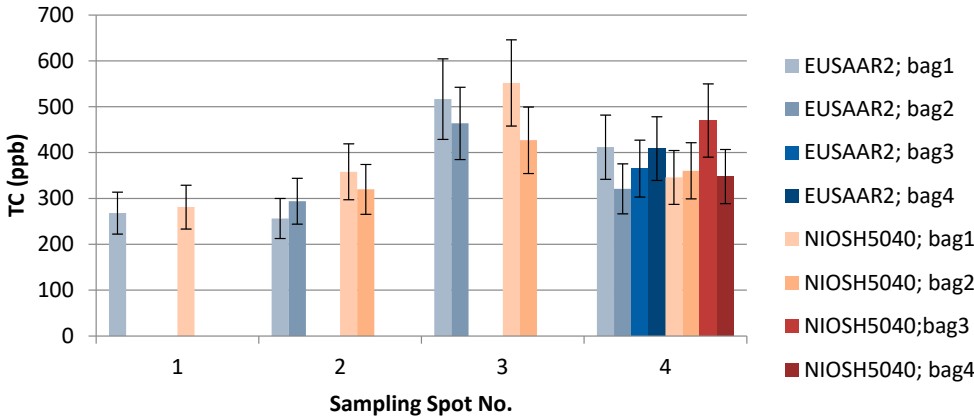

**Figure 7.** The effect of EUSAAR2 and NIOSH5040 analysis protocols on the detected TC.

Differences between the sampling spots are discussed later, when the results of all laboratories are compared.

For University of Florence, the results of the OC and EC measurements as μgC on the total surface of the filter are reported in Table 1, together with the volume of the filtered sample, the blank values calculated for each filter, and the final results in ppb (blank values are subtracted for ppb calculation). The error related to the measurement is 6%, and about 20% for OC and EC, respectively. This error originated only from the measurement as for each sampling location only one sample was collected; also, as the deposited material on the filter was not homogeneously distributed on the filter surface, all of the filter was analyzed. Therefore, the reported error was underestimated with respect to those reported by TU-Wien and FMI. Figure 8 reports the OC, EC, and TC concentrations in each sampling location.

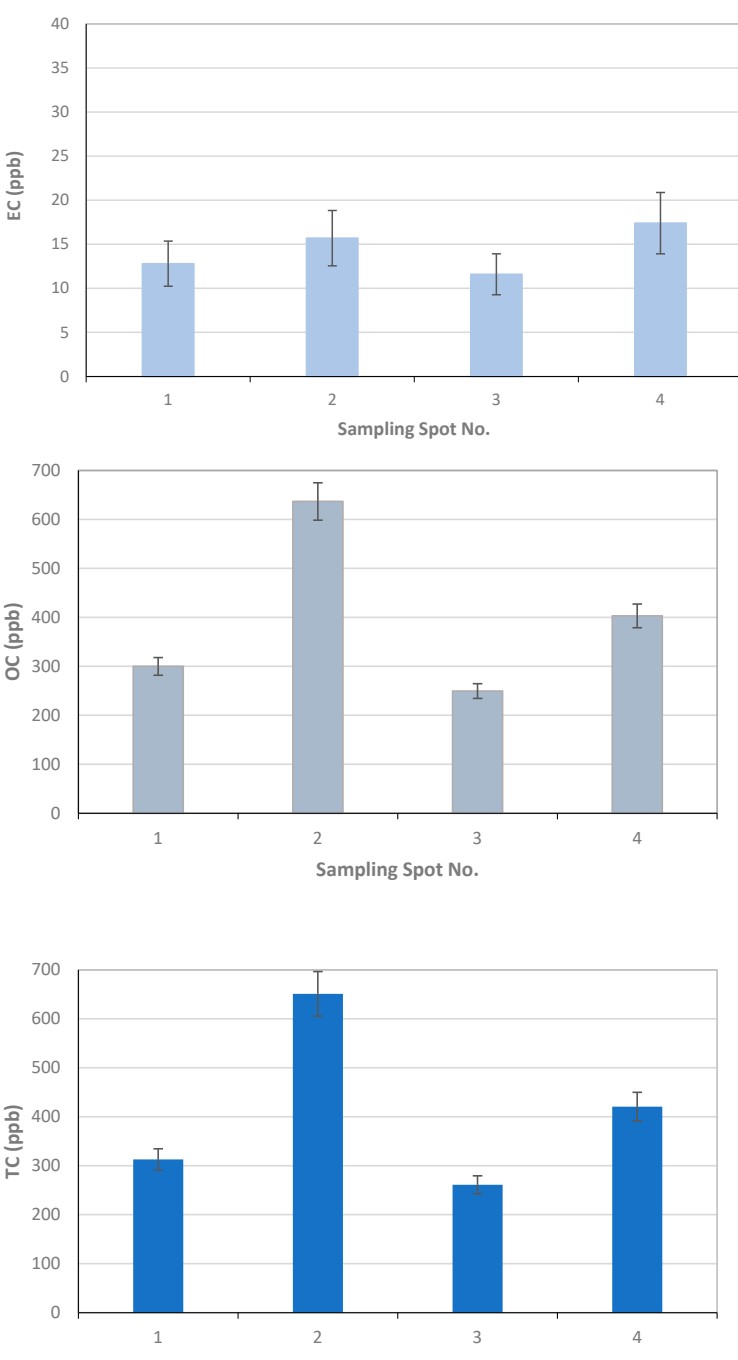

**Figure 8.** Results of University of Florence for EC, OC, and TC concentrations and associated uncertainties measured in the four sampling locations S1–S4.

For the Finnish Meteorological Institute, the results of the EC, OC, and TC analysis as mass of carbon fraction in filter samples as [$\mu g/cm^2$] are reported in Table 3, together with the volume of the filtered snow sample in [ml]. Each filter was used to analyze three replicates marked as a, b, and c. The concentration of the EC, OC, and TC per volume of snow in [ppb] is calculated using the volume of melted snow and the amount of carbon compound in the analyzed filter piece multiplied by the filter area (Figure 9).

**Table 3.** The EC, OC, and TC results of Finnish Meteorological Institute. The analysis uncertainty is $\pm 0.2$ $\mu$gC (+5% relative error for higher loaded samples).

| Sample ID | Volume of Melted Snow (mL) | Mass of EC Fraction in Filter Sample ($\pm$unc) $\mu g/cm^2$ | Mass of OC Fraction in Filter Sample $\mu g/cm^2$ | Mass of TC in Filter Sample $\mu g/cm^2$ | EC per Volume Snow (ppb) | OC per Volume Snow (ppb) | TC per Volume Snow (ppb) |
|---|---|---|---|---|---|---|---|
| S1aFMI | 350 | 0.47 ($\pm$0.12) | 6.77 | 7.24 | 12.1 | 175.5 | 187.7 |
| S1bFMI | 350 | 0.65 ($\pm$0.13) | 8.78 | 9.43 | 16.9 | 227.6 | 244.5 |
| S1cFMI | 350 | 0.58 ($\pm$0.13) | 9.11 | 9.69 | 15.0 | 236.2 | 251.2 |
| S2aFMI | 670 | 1.27 ($\pm$0.16) | 11.75 | 13.02 | 17.2 | 159.1 | 176.3 |
| S2bFMI | 670 | 1.01 ($\pm$0.16) | 10.18 | 11.21 | 14.0 | 137.9 | 151.8 |
| S2cFMI | 670 | 1.03 ($\pm$0.15) | 9.47 | 10.48 | 13.7 | 128.3 | 141.9 |
| S3aFMI | 430 | 0.73 ($\pm$0.14) | 13.43 | 14.16 | 15.4 | 283.4 | 298.8 |
| S3bFMI | 430 | 0.97 ($\pm$0.15) | 12.2 | 13.17 | 20.5 | 257.5 | 277.9 |
| S3cFMI | 430 | 0.3 ($\pm$0.11) | 12.83 | 13.12 | 6.3 | 270.8 | 276.9 |
| S4aFMI | 580 | 1.1 ($\pm$0.16) | 26.7 | 27.8 | 17.2 | 417.7 | 435.0 |
| S4bFMI | 580 | 0.9 ($\pm$0.14) | 14.38 | 15.27 | 14.1 | 225.0 | 238.9 |
| S4cFMI | 580 | 1.0 ($\pm$0.15) | 16.13 | 17.13 | 15.6 | 252.4 | 268.0 |

### 3.5. Comparison of the Results from the Three Laboratories

The results of the thermal-optical analysis from the three participating laboratories are presented in Figure 10. The difference in meter scale, i.e., between sampling spots S1, S2, S3, and S4, was minimal for EC (Tables 1–3, Figure 9). For S1, EC varied 12–17 ppb; for S2, 14–19 ppb; for S3, 6–28 ppb; and for S4, 14–28 ppb. In the case of TU Wien, EC values for S1–S4 were 15–31 ppb; for the University of Florence 12–18 ppb; and for the Finnish Meteorological Institute, 6–21 ppb. The median measured EC concentration was 16.3 ppb, and the average was 18.3 ppb, with a standard deviation of 5.9 ppb. All the detected EC concentrations represent clean snow. OC varied for S1 between 175 and 300 ppb; for S2, between 128 and 279 ppb (except one potential outlier of 639 ppb); for S3, between 250 and 489 ppb; and for S4, between 225 and 417 ppb. In the case of TU Wien, the OC detected for sample spots S1–S4 was between 237 and 489 ppb; in the case of the University of Florence, between 249 and 636 ppb; and in the case of the Finnish Meteorological Institute, between 137 and 437 ppb. Hence, for OC concentrations, the difference in meter scale was larger than for EC (Tables 1–3, Figure 10), and the detected carbon contents were determined by the fraction of OC. The observed variability in the measured carbon compounds (Figure 10) can be due to natural surface concentration variability or an inhomogeneous filter. Inhomogeneity in filter loading can be suspected in the case of the sample S3c, analyzed by the Finnish Meteorological Institute, where only 6.3 ppb of EC was detected, while the two other filter pieces cut from the same filter contained 15 and 21 ppb EC. One possible outlier was detected in the sampling spot S2 result for OC, analyzed by the University of Florence. In that case, the OC concentration was 639 ppb, while otherwise all the OC concentrations remained under 489 ppb. The reason for this outlier remains unclear, and therefore it may be due to contamination but may also due to organic carbon in the sample.

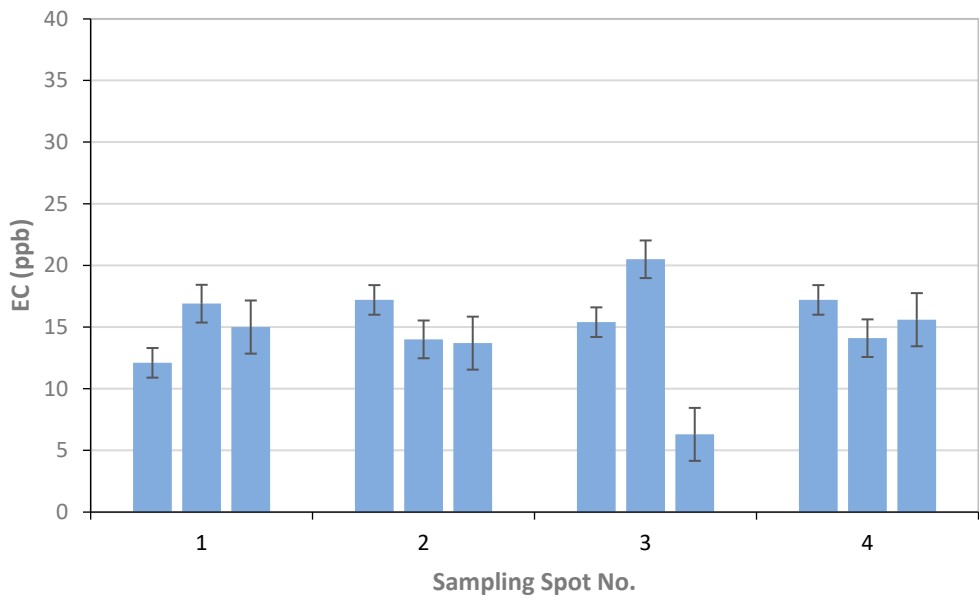

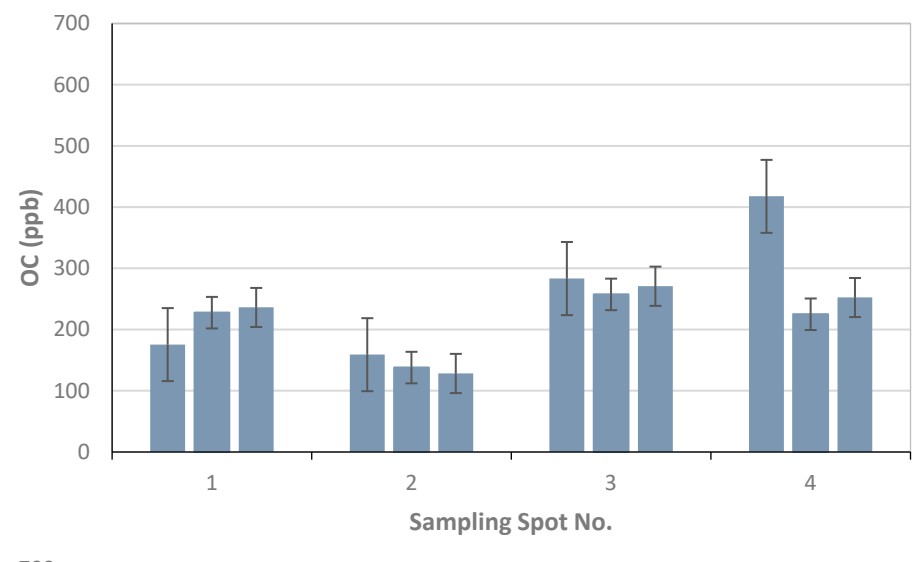

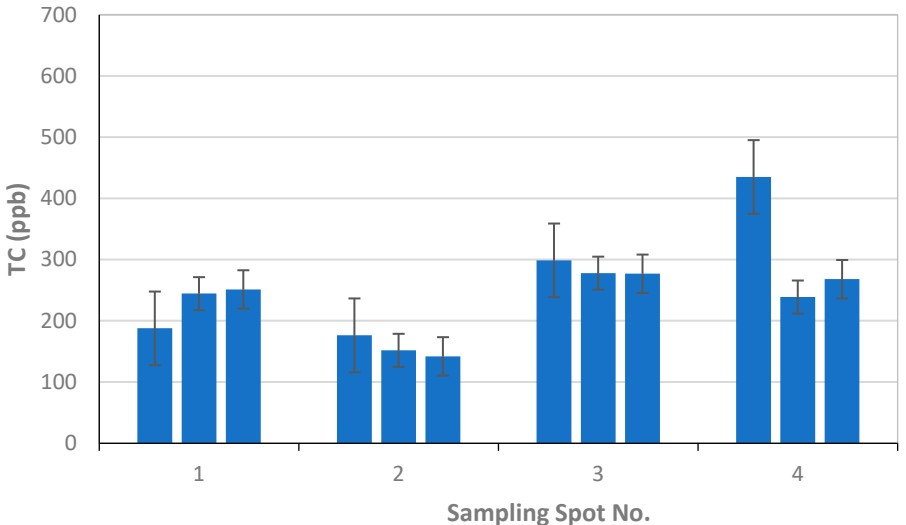

**Figure 9.** Finnish Meteorological Institute results for EC, OC, and TC concentrations using three replicates per sample filter from the four sampling spots 1–4 and associated standard error bars calculated from the three replicates of one sample.

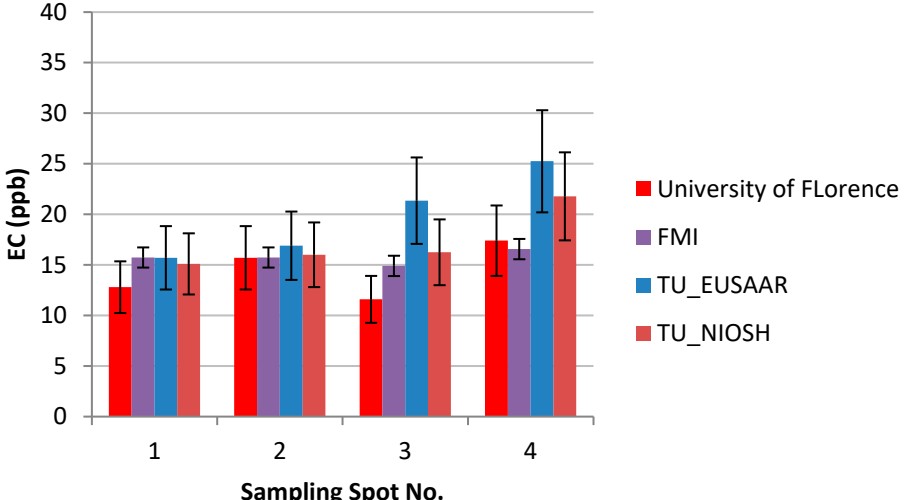

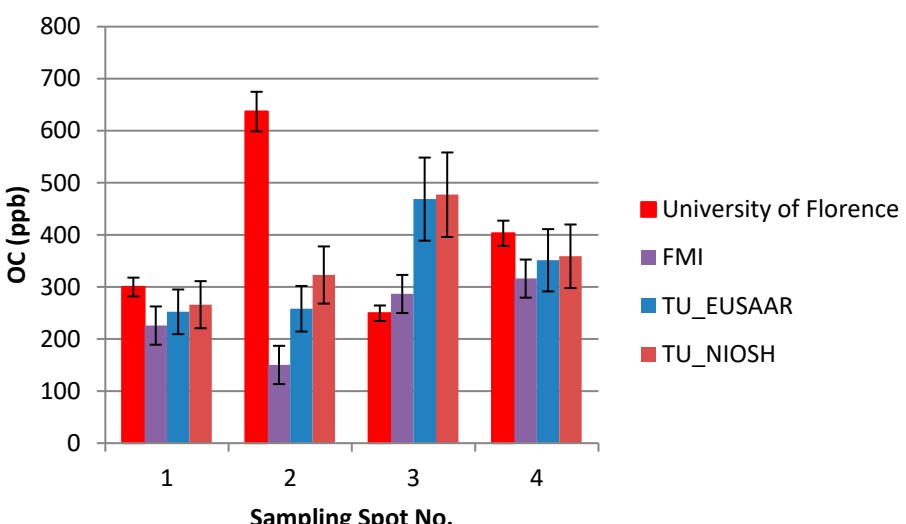

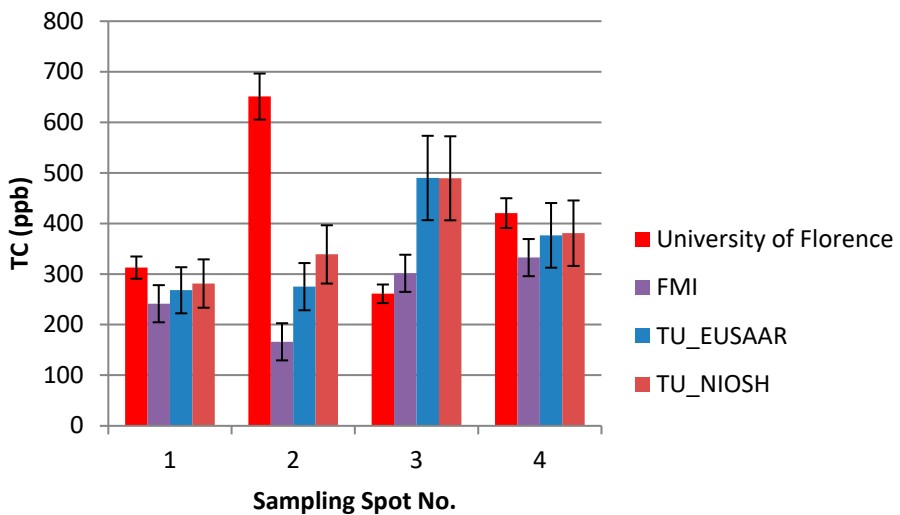

**Figure 10.** The results of the laboratory analysis from the three participating laboratories.

Overall, the carbon content differences the in meter scale appeared smaller than the difference between the three laboratories. From this it can be concluded that the observed concentrations can be considered representative for the region at the end of February 2018, since the TC, OC, and EC values of the four nearby sampling spots did not show meter-scale horizontal variability in surface snow.

## 4. Recommendations for the Key Parameters an Intercomparison Participant Should Be Asked For

The key parameters that intercomparison participants should be asked for are first listed here (Sections 4.1–4.5). Thereafter, the overall conclusions from the intercomparison experiment are presented (Section 4.6).

### 4.1. Snow Sampling

- What are the sampling procedures? For intercomparison purposes, the most critical question is how the sampling is designed to gain as homogenous samples as possible for each intercomparison laboratory (e.g., using the sampling procedure presented in this paper). Sampling procedures include the number of sample locations; what environment the sampling locations represent (e.g., arctic, mountain, urban); how the sampling locations are chosen (what the distance between the locations is, whether the locations are in a line, in grid, or located randomly); how many parallel samples are collected to represent one sampling location; the amount of snow needed for one sample (may require a pre-visit to the site prior to the inter-comparison; the amount of snow needed depends on, e.g., the impurities and density of snow; in the filter method, if too many impurities are in the filter, the analysis can be oversaturated, and in too small amounts, the detection limit is not exceeded); defining samples to represent surface snow, whole snow pack as one sample, or using snow pit samples collected by snow layer or fixed increments; and description of the sampling devices and their cleaning before and during the sampling.

### 4.2. Sample Transport, Storage, and Melting

- How are the snow samples stored and transported during the sampling and from the sampling site to the laboratory? How are the samples stored in the laboratory? For example, are the snow samples transported to the laboratory as snow, or are they allowed to melt and then filtered right after? Storage refers to how the snow samples are stored during the sampling and whether they are kept frozen after sampling, how the melting of the snow sample is conducted, how the sample filters are stored and transported to the laboratory, and whether traveling blank filters are used and what the procedures are regarding their transport and storage.

### 4.3. Filtering

- What are the filtering procedures? This includes the used filter type, how the travel blank filters are treated during filtering, description of the filtering devices, how the filtering devices are cleaned before and during the filtering process, how the filters are dried after filtering, how the filters are stored before, during, and after sampling, and how the filters are transported to the sampling site and to the laboratory.

### 4.4. OCEC Analysis

- Blank filter: is a laboratory blank filter analyzed before analysis of the intercomparison samples?
- Sucrose test: how often is a sucrose test performed? Is the sucrose test performed before analyzing the intercomparison samples?
- Replicates: how many replicates are used to represent one filter sample?
- Analysis protocols: what OCEC analysis protocol is used?
- Split point: how is the split point determined?

*4.5. Data Analysis*

- Do the results follow a normal distribution or not? If this is not known, the median value should be given instead of the average.
- What are the reasons for outliers?

*4.6. Conclusions from the Intercomparison Experiment*

The estimation of the overall uncertainty requires all the steps of sampling, filtering, and analysis and cannot be reached by measuring the same filter sample in different laboratories. Therefore, intercomparisons which include all the steps are needed. Such intercomparisons can be performed using the procedures of the participating laboratories, but require a detailed documentation of the procedures and protocols used. Low carbon concentrations can be detected using the shorter NIOSH5040 protocol. Our studies suggest that the NIOSH5040 protocol might be equally suitable for detecting larger carbon concentrations in snow, and this could be studied in the future in more detail. For the location of this experiment, the origin of OC could be studied further. Previously [6], in Sodankylä, Finland, OC was found to be due to organic matter (tree litter), while in an urban area it can be due to air pollution.

In summary, to reduce the uncertainty in future intercomparisons of water-insoluble carbonaceous particles in snow, we recommend the procedure to include the following critical steps: to avoid contamination it is necessary to use sterile equipment and clean water (e.g., MilliQ) in rinsing; pre-sampling of snow is needed to estimate the correct amount of snow to be collected for the intercomparison purpose, as the impurity concentrations need to be above the detection limit and under the saturation level of the analyzer; snow needs to be collected for the participating laboratories in such a way that the results to be compared against each other represent the same snow (e.g., using the intercomparison sampling method presented here); the minimum amount of snow samples to be collected to represent one location is four, and the minimum number of participating laboratories is three; each participating laboratory should use at least one traveling reference filter and one blank filter in their analysis, and perform a sucrose tests before sample analysis; the protocol files of each laboratory should be compared to be exactly the same for the traveling reference, blank, sucrose test, and sample filters.

The results of this experiment demonstrated that the intercomparison methods used in this research for sampling, filtering, and laboratory analysis were suitable for detecting water insoluble total, organic, and elemental carbon in snow. To broaden the scientific outcome of this study, similar investigations should be repeated in the future with larger materials in polar and mountain environments and should include various snow conditions of accumulation and melt periods representing the six global snow classes [16].

**Author Contributions:** Writing—original draft preparation, O.M.; writing—review and editing, A.K.-G., S.B., D.K. and W.S.; Sonnblick intercomparison participants, O.M., A.K.-G., S.B. and W.S.; Snow sampling and filtering, O.M. and S.B.; filtering of TU-Wien samples, D.K.; OCEC analysis, O.M., D.K. and G.C.; OCEC analysis quality control and supervision, M.A. and G.C.; Sonnblick Workshop organization, W.S. All authors have read and agreed to the published version of the manuscript.

**Funding:** This intercomparison experiment was funded by EU ESSEM COST Action ES1404 Harmosnow. O.M. was funded by the Academy of Finland (ACCC No. 337552 and BBrCAC No. 341271) and Ministry for Foreign Affairs of Finland (IBA-project No. PC0TQ4BT-20).

**Institutional Review Board Statement:** "Not applicable" for studies not involving humans or animals.

**Data Availability Statement:** Data are mostly included in this article or else available on request via personal communication.

**Acknowledgments:** Steven Warren, University of Washington, USA, is gratefully acknowledged for his guidance in Norway to OM regarding snow sampling for intercomparisons. Ali Nadir Arslan, FMI, is acknowledged for his work and efforts for ES1404. NordSnowNet project of the Nordregio and the Nordic Council of Ministers Arctic Co-operation Programme, as well as the Ministry for Foreign Affairs of Finland projects of IBA-BCDUST and IBA-Permafrost, are gratefully acknowledged.

**Conflicts of Interest:** The authors declare no conflict of interest. The funders had no role in the design of the study; in the collection, analyses, or interpretation of data; in the writing of the manuscript; or in the decision to publish the results.

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
