# Peer review of "Intercomparison Experiment of Water-Insoluble Carbonaceous Particles in Snow in a High-Mountain Environment (1598 m a.s.l.)"

_geosciences, doi:10.3390/geosciences12050197_

Round 1

Reviewer 1 Report

Comments on the manuscript “Intercomparison experiment of water-insoluble carbonaceous particles in the snow in a high-mountain environment (1598 m a.s.l)” by Meinander, O. et al.

General:

Precise evaluation of OC and EC's concentration in the snow is essential for the accurate estimation of climate change. The authors have conducted an intercomparison experiment using the same samples, and they have shown that their methods are applicable for future research and monitoring of carbonaceous particles in snow. The objective of their study is quite essential for the research on climate change and air quality, and the method is well described in the manuscript. The manuscript seems to be within the scope of this journal, and their results and suggestion appears to be quite valuable for the research community and is worth publication after minor corrections.

Detailed:

Page 5, Figure 1: Gridlines for latitude/longitude or scale for the unit distance should be added in the left panel of Figure 1.

Page 6: “S1”, “S2”, “S3”, and “S4” are used without explanation. If they are the abbreviation of “Spot 1”, “Spot 2”, “Spot 3”, and “Spot 4”, the authors should add an explanation in the revised manuscript.

Page 6, Figure 3: Distances among sampling locations are written in the text (Page 6), but the scale is not shown in Figure 3. The authors should indicate the distance of each spot.

Reviewer 2 Report

The manuscript is interesting and the motivation clear and mandatory: the harmoization of samöling, sample preparation and laboratory analysis. The reference here is limited and only for few regions, like Svlabard, northern Finland and US high mountain area exist some useful instruction. The strategy is very clever to sample at the same place snow probes and analysis is at different place based on a similar/same instruction. The entire strategy is well present and show the riscs and the advantage. The complete paper is well structured with good reading and a message. My only concern is here, that the scientific outcome is limited. As technical note should  be this manuscript announced. Because of up to now aclear instruction for probing and analysing is missing, this kind of paper is extremely important. The question is, what means the difeerences in the resulty with slightly different analysis procedure. A final recommendation and a clear describe strategy / procedure to reduce the uncertainty is still missing and should be add.
